

# New digital anatomical data of *Keichousaurus hui* (Reptilia: Sauropterygia) and its phylogenetic implication

Jiayu Xu[1,*], Yu Guo[2,*], Yucong Ma[1], Wei Wang[3], Long Cheng[4] and Fenglu Han[1]

[1] School of Earth Science, China University of Geosciences, Hubei, Wuhan, China
[2] The Geological Museum of China, Beijing, China
[3] Institute of Vertebrate Paleontology and Paleoanthropology, Chinese Academy of Sciences, Beijing, China
[4] Hubei Key Laboratory of Paleontology and Geological Environment Evolution, Wuhan Centre of China Geological Survey, Hubei, Wuhan, China
[*] These authors contributed equally to this work.

## ABSTRACT

Three complete skulls of *Keichousaurus hui* from the Middle Triassic Xingyi Fauna of southwestern China were examined using high-resolution computed tomography (CT) scanning. The CT images allow a few refinements and supplements in cranial anatomy. Some previously ambiguous anatomical characters were identified, including the presence of an L-shaped ectopterygoid that extends from the lateral side of the pterygoid and bends ventrally, the wedge-shaped posterolateral process of the frontal, the trapezoidal pterygoid for articulating with the palatine, and a rodlike basioccipital tuber that extends posterolaterally. These new features provide new detailed anatomical information for taxonomy. The new phylogenetic analysis of Sauropterygiformes places *Keichousaurus* as an eosauropterygian that is more basal than the monophyly which includes Nothosauridae and Pistosauroidea. Moreover, the result also suggests that *Keichousaurus* is more closely related to Chinese pachypleurosaurs-like eosauropterygians than to European pachypleurosaurs and more derived than other Chinese pachypleurosaurs-like forms.

## INTRODUCTION

*Keichousaurus hui* is a small marine reptile, normally less than 0.5 m in body length, that inhabited the eastern Tethys during the Ladinian stage of the Middle Triassic (*Lin & Rieppel, 1998*). This taxon was first discovered in the Zhuganpo Member of Falang Formation in Dingxiao Village, Xingyi City, Guizhou Province of China (*Young, 1958*). Subsequently, abundant materials have been collected from neighboring localities spanning Guizhou and Yunnan provinces (*Li & Jin, 2003*; *Ma et al., 2013*). Several aspects of this taxon have been thoroughly investigated including the analysis of the fossil-bearing strata and corresponding paleoenvironment (*Young, 1965*; *Chen, 1985*; *Wang, 1996*; *Yang, 1997*; *Wang, Kang & Wang, 1998*; *Li & Jin, 2003*; *Sun, Hao & Jiang, 2005*; *Ma et al., 2013*; *Hu,*

Corresponding author
Fenglu Han, hanfl@cug.edu.cn

*Xie & Yin, 2018*), osteological anatomy (*Cheng & Pan, 1999*; *Holmes, Cheng & Wu, 2008*; *Liao et al., 2021*), ontogenetic stages (*Fu et al., 2013*; *Qin, Yu & Luo, 2014*), and sexual dimorphism (*Cheng et al., 2009*; *Xue et al., 2013*). However, the internal anatomy of the skull remains poorly understood due to fossil preservation.

In recent years, computed tomography (CT) has been widely used in studying the osteological morphology of vertebrate fossils, including the Triassic marine reptiles (*e.g.*, *Neenan et al., 2015*; *Čerňanský et al., 2018*; *Wang et al., 2019*; *Yin, Zhou & Lu, 2021*). Here, the cranial morphology of three *Keichousaurus* skulls is studied based on CT scanning. By comparing with the interpretations of the skulls of *Keichousaurus* presented in *Lin & Rieppel (1998)* and *Holmes, Cheng & Wu (2008)*, and other previous work, new features on the internal anatomy of the skull elements have been identified in more comprehensive views. Some characteristics in *Keichousaurus* are supplemented and revised, mainly including the morphological features of the dentition, the hyobranchium, the ectopterygoid and other cranial elements. A detailed description of these new features is provided, with comparisons to those of other pachypleurosaurs. This discovery is significant for refining the phylogenetic position of *Keichousaurus* in eosauropterygians.

## MATERIALS & METHODS

Three well-preserved specimens of *Keichousaurus hui* (Fig. S1) are employed for this study.

These fossils were collected from the thin limestone of the Middle Triassic Zhuganpo Member of Falang Formation at Mayigou locality, Fuyuan County, Yunnan Province, southwestern China (Fig. S2). These specimens are housed in the paleontological collection of China University of Geosciences (Wuhan) (CUGW) with specimen numbers of CUGW VH007, CUGW VH009, and CUGW VH017, respectively. All three specimens have not been prepared and have retained their original buried state.

The skulls of *Keichousaurus* were scanned by the micro-CT scanners at the Institute of Vertebrate Paleontology and Paleoanthropology, Chinese Academy of Sciences (IVPP) and Yinghua Inspection and Testing (Shanghai) Co., Ltd. Among these specimens, CUGW VH009 was scanned using the 225 kV micro-CT at IVPP, with a voltage of 180 kV, a current of 100 μA, and a resolution of 21.96 μm per pixel. CUGW VH007 and CUGW VH017 were scanned in the industrial micro-nanometer CT v|tome|x m of Yinghua Inspection and Testing (Shanghai) Co., Ltd. with a voltage of 140 kV, the current of 100 μA, and the resolution of 10.71 μm per pixel. The CT data of the three specimens were reconstructed using Mimics 19.0, and the resulting models were rendered and visualized (Figs. 1 and 2). The bony elements of CUGW VH009 were best reconstructed, separated, rendered, and illustrated in different colors (Fig. 1). The description and comparisons in this study are primarily based on this specimen. Three reconstructions of the CT data have been uploaded to MorphoSource (see Data S1).

Ontogenetic variation and sexual dimorphism in *Keichousaurus* have been well-studied in previous researches (*Lin & Rieppel, 1998*; *Cheng et al., 2009*; *Qin, Yu & Luo, 2014*).

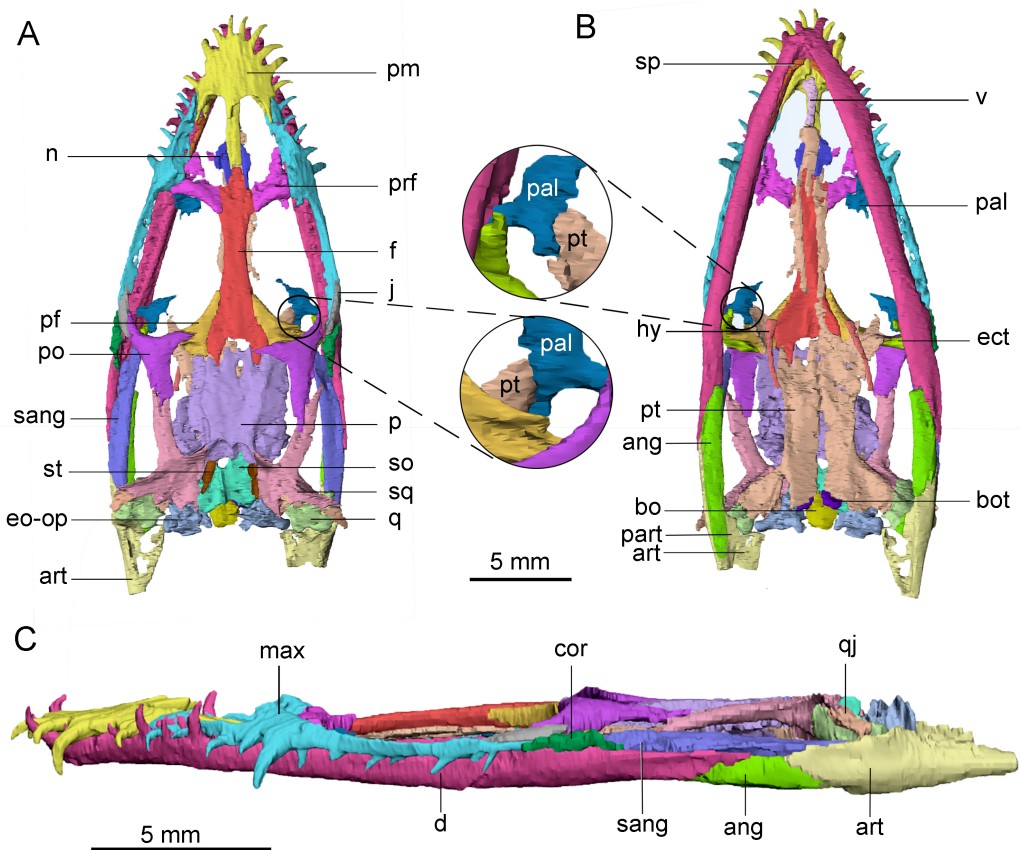

**Figure 1** **3D reconstruction of the skull of *Keichousaurus hui* (CUGW VH009). The colors denote different bones.** (A) Dorsal view; (B) ventral view; (C) left lateral view. The circle shows that pterygoid contacts with palatine. Abbreviations: ang, angular; art, articular; bo, basioccipital; bot, basioccipital tuber; cor, coronoid; d, dentary; ect, ectopterygoid; eo-op, exoccipital-opisthotic; f, frontal; hy, hyobranchium; j, jugal; max, maxilla; n, nasal; p, parietal; pal, palatine; part, prearticular; pf, postfrontal; pm, premaxilla; po, postorbital; prf, prefrontal; pt, pterygoid; q, quadrate; qj, quadratojugal; sang, surangular; so, supraoccipital; sp, splenial; sq, squamosal; st, supratemporal; v, vomer.

According to body length, CUGW VH007 and CUGW VH009 are possibly to be in the adult stage, while CUGW VH017 is in the sub-adult stage (Table 1). CUGW VH007 could be identified as a male individual based on a humerus to femur ratio of approximately 1.3 and the expansion of humeral condyles, whereas CUGW VH009 and CUGW VH017 are identified as female according to the humerus to femur ratio of approximately 1 and the smooth distal end of the humerus (Table 1).

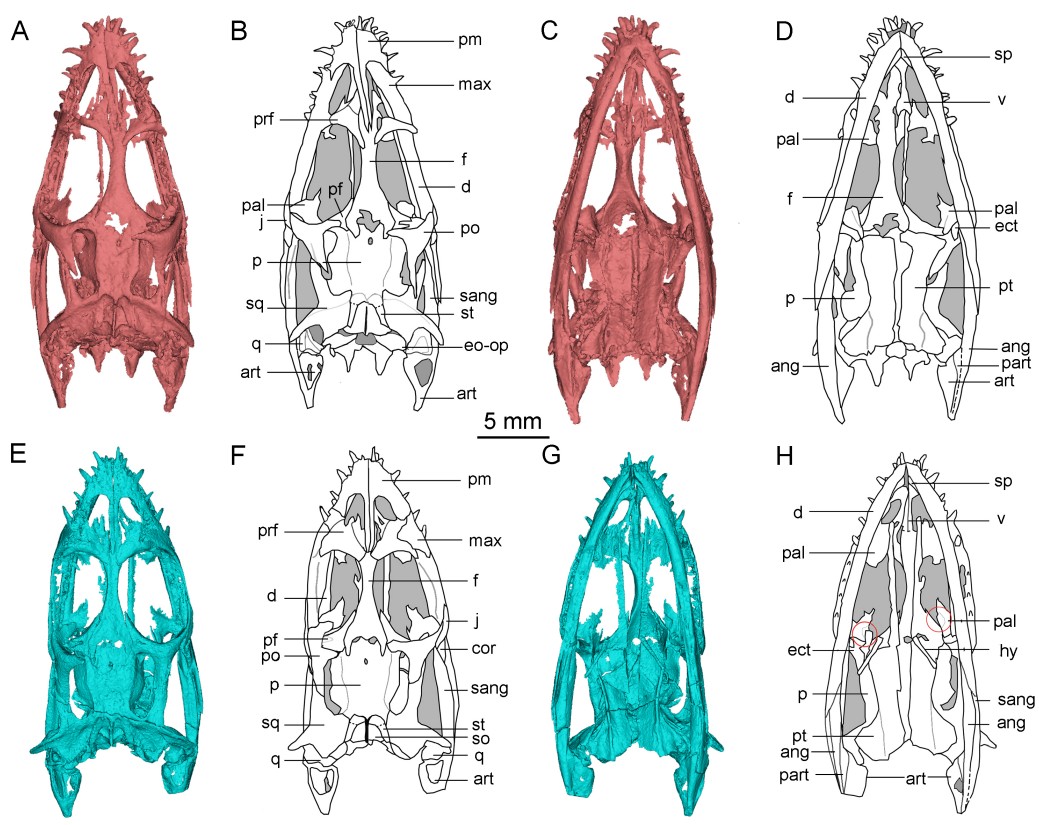

**Figure 2** **3D reconstruction and outline drawing of the skulls of *Keichousaurus hui*.** (A–D) CUGW VH007; (A, B) dorsal view; (C, D) ventral view. (E–H) CUGW VH017; (E, F) dorsal view; (G, H) ventral view. The circle is where the pterygoid connects the palatine. The dashed line represents the uncertain suture. The shadow region denotes cavities and/or unreconstructed bones. Abbreviations as in Fig. 1.

## RESULTS

### Systematic paleontology

Diapsida *Osborn, 1903*
Sauropterygia *Owen, 1860*
Eosauropterygia *Rieppel, 1994*
*Keichousaurus Young, 1958*
*Keichousaurus hui Young, 1958*

### Description and comparison

All skulls are well preserved and are compressed dorsoventrally. The skull of *Keichousaurus* is generally wedge-shaped in dorsal view and the widest part of the skull is located in the posterior orbital region (Figs. 1, 2A, 2B, 2E and 2F). The skull length of two adult individuals (CUGW VH007 and CUGW VH009) measures 24.7 mm and 24.3 mm, respectively. The external nare is long and narrow with a subtriangular outline. The orbit is pronounced,

**Table 1  Measurement record of *Keichousaurus hui*.**

| Specimens | Skull length (mm) | Humerus (mm) | | Femur (mm) | | Snout-vent length (mm) | Stage | Gender |
|---|---|---|---|---|---|---|---|---|
| | | Left | Right | Left | Right | | | |
| CUGW VH007 | 24.7 | 22.8 | 22.7 | 17.2 | 17.0 | 150.4 | Adult | Male |
| CUGW VH009 | 24.3 | 15.8 | 16.1 | 15.6 | 16.0 | 154.2 | Adult | Female |
| CUGW VH017 | 19.8 | 11.0 | 12.6 | 10.5 | 12.1 | 114.7 | Subadult | Female |

comprising approximately 25 percent of the skull length. The preorbital region is slightly longer than the postorbital region. The supratemporal fenestra is elongated and about 70% of the orbital length, and the posteromedial margin of the skull roof is concave. Based on previous studies (*e.g., Lin & Rieppel, 1998*; *Holmes, Cheng & Wu, 2008*; *Liao et al., 2021*), this article provides novel and revised anatomical interpretations.

## Frontal

The frontal is elongated and flat, forming the medial edge of the orbit. It articulates with the nasal anteriorly and the prefrontal anterolaterally. The posterior end is bifurcated and covered on the parietal (Figs. 1A, 2A, 2B, 2E and 2F). The posterolateral process gradually tapers and inserts into the depression in the anterior region of the parietal, similar to the condition in *Qianxisaurus* (*Cheng et al., 2012*), *Dianmeisaurus* (*Shang & Li, 2015*) and *Dawazisaurus* (*Cheng et al., 2016*). This contrasts with the previous description, which suggested that the posterolateral process is arc-shaped (*Lin & Rieppel, 1998*; *Holmes, Cheng & Wu, 2008*) (Figs. 3A and 3B). The frontal is excluded from the supratemporal fenestra by the postfrontal and parietal.

The posterolateral processes of the frontal in *Keichousaurus* are nearly parallel, which are similar to those of *Diandongsaurus* (*Shang, Wu & Li, 2011*; *Sato et al., 2014*) and *Dianopachysaurus* (*Liu et al., 2011*). However, the processes are different from those of *Honghesaurus* (*Xu et al., 2022*), *Wumengosaurus* (*Jiang et al., 2008*; *Wu et al., 2011*), and European pachypleurosaurs, such as *Anarosaurus* (*Klein, 2009*), *Serpianosaurus* (*Rieppel, 1989*) and *Neusticosaurus* (*Sander, 1989*), in which the posterior end of the frontals bifurcates into two processes with a significant angle. The frontal in *Keichousaurus* differs from that of *Panzhousaurus*, as *Panzhousaurus* lacks a posterolateral process (*Jiang et al., 2018*; *Lin et al., 2021*).

## Basioccipital tuber

A pair of short, columnar bony processes extending posterolaterally on the ventral side of the basioccipital are identified as basioccipital tubera (Fig. 4). These tubera extend from the anterior end of the ventral side of the basioccipital and lie on the posterodorsal side of the pterygoid. The well-preserved left tubercle is short, cylindrical, and extends posterolaterally, forming an angle of about 60° with the central axis of the skull. The distal end of the tubercle is smooth and rounded. The right tubercle appears less intact due to the damage in the occipital region.

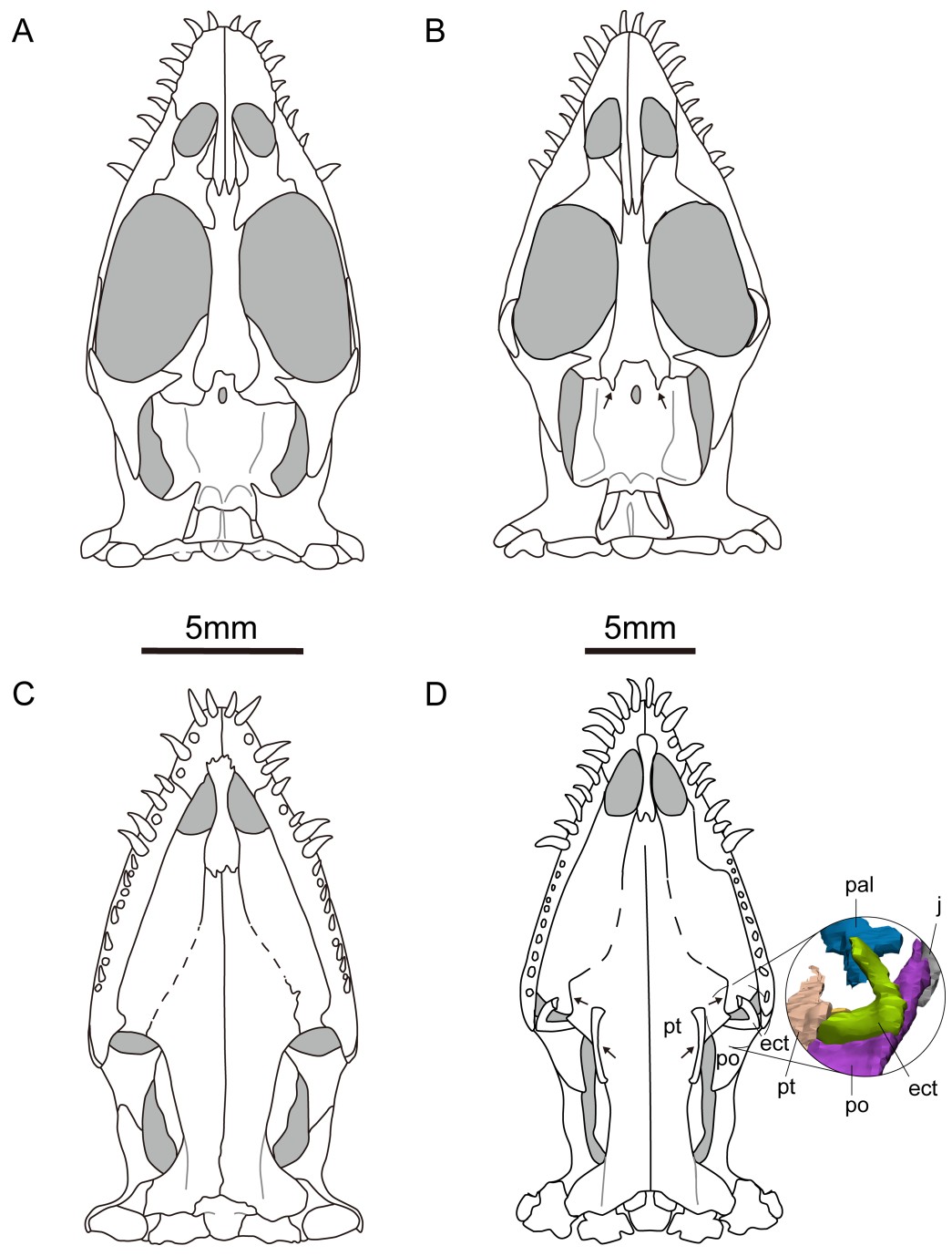

**Figure 3** **Reconstruction of the skull of *Keichousaurus hui*.** (A, B) Dorsal view; (A) from *Holmes, Cheng & Wu (2008)*; (B) according to our study. (C, D) Ventral view; (C) from *Holmes, Cheng & Wu (2008)*; (D) according to our study. Arrows point differences between our study and *Holmes, Cheng & Wu (2008)*. The circle shows the ectopterygoid bone. The shadow area denotes no bone. Abbreviations as in Fig. 1.

The basioccipital tuber is similar to those of nothosaurs and plesiosaurs (*Rieppel, 1994*; *Storrs & Taylor, 1996*), but it has not previously been reported in pachypleurosaurs. This

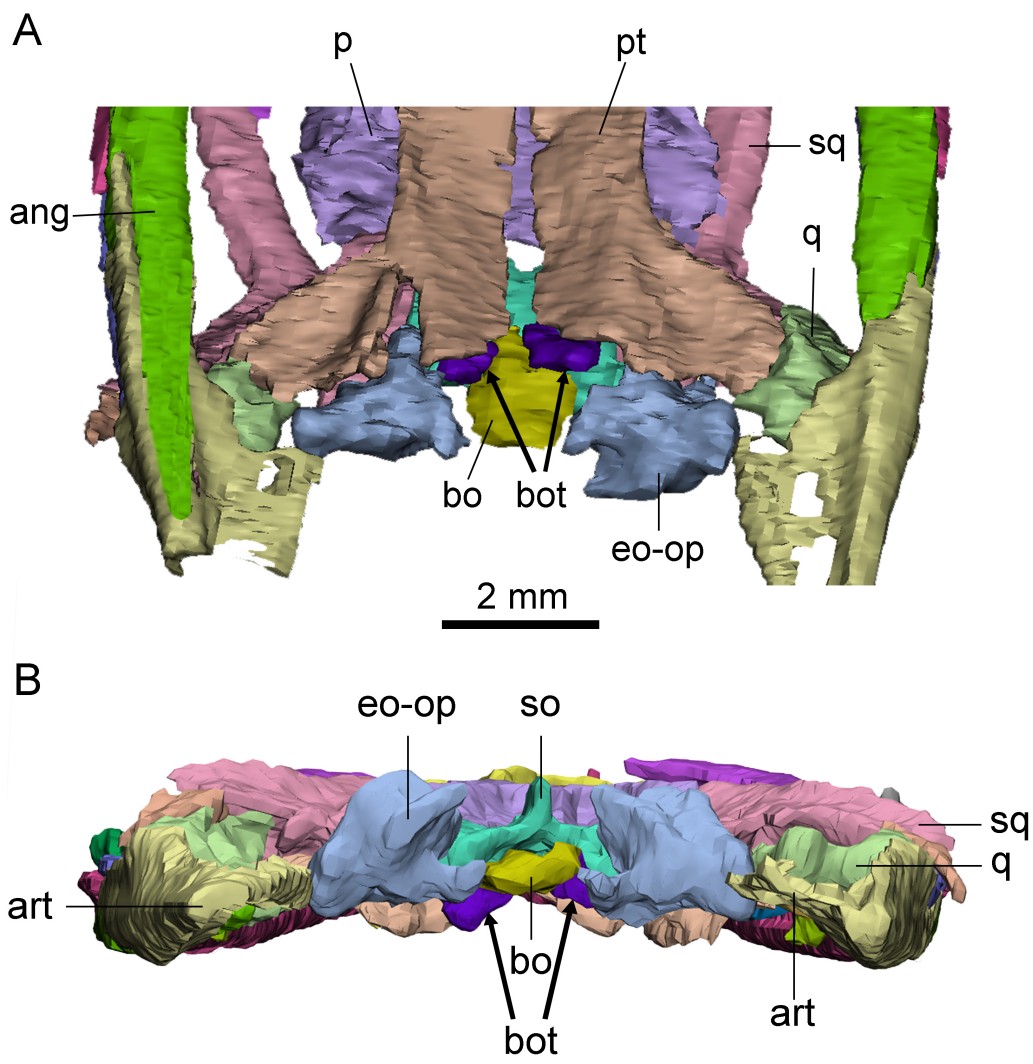

**Figure 4  Posterior skull of *Keichousaurus hui* (CUGW VH009).** (A) Ventral view; (B) occipital view. Arrows denote basioccipital tuber. Abbreviations as in Fig. 1.

new finding confirms the presence of basioccipital tuber in *Keichousaurus* for the first time, suggesting that it may also be present in other early eosauropterygians.

## Pterygoid and palatine

The pterygoid of *Keichousaurus* is long and strip-like, occupying the central position on the ventral side of the skull. In CUGW VH009, the lateral edge of the pterygoid narrows slightly and contacts with the depressed palatine (Fig. 1B). In CUGW VH007, the pterygoid inserts into the groove of the palatine. In CUGW VH017, the pterygoids on both sides do not fully contact the palatine due to preservation, but based on the morphology of the palatine and its articular surface of the pterygoid, as well as the partially preserved lateral edge of the pterygoid bone, it can be inferred that the middle portion at the lateral edge of the pterygoid exhibits a trapezoidal expansion. The anterior portion of the trapezoidal

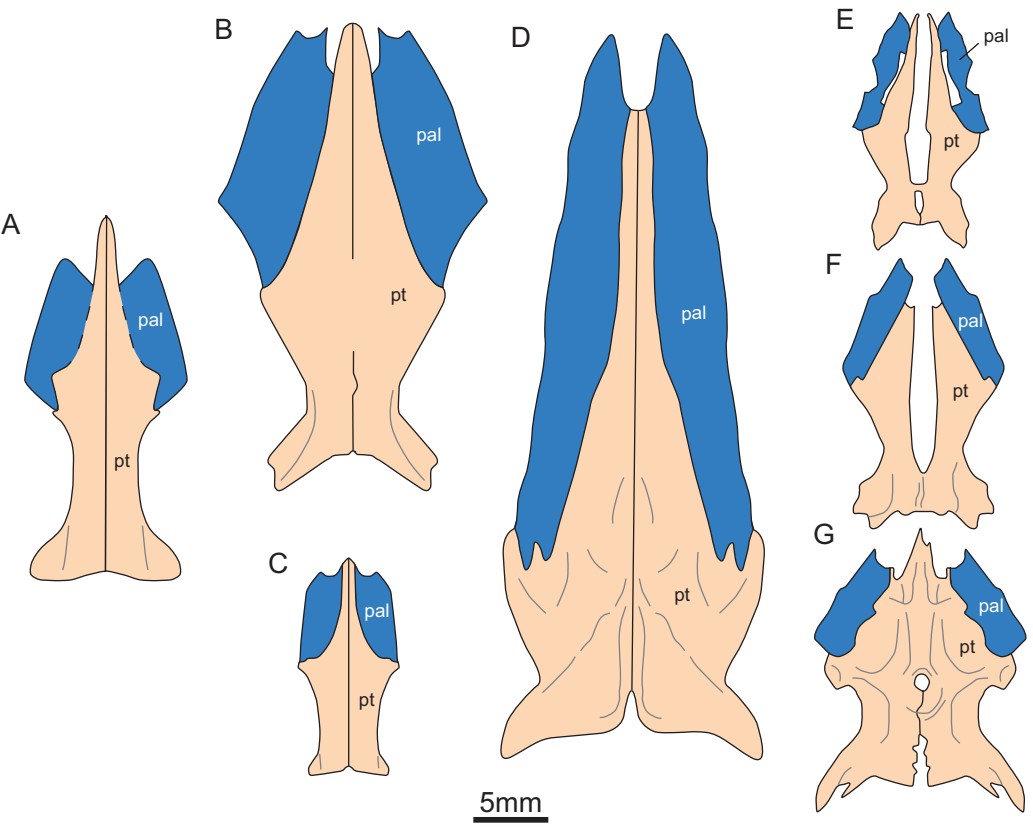

**Figure 5  Reconstruction of pterygoid and palatine of eosauroptetygians.** (A) *Keichousaurus hui*; (B) *Serpianosaurus mirigiolensis*; (C) *Neusticosaurus peyeri*; (D) *Wumengosaurus delicatomandibularis*; (E) *Panzhousaurus rotundirostris*; (F) *Dianmeisaurus gracilis*; (G) *Diandongsaurus acutidentatus*. Abbreviations as in Fig. 1.

expansion forms an obtuse angle that inserts into the dorsal concave surface of the palatine, whereas the posterior region forms a small triangular buttress for contacting the distal end of the palatine (Figs. 2G, 2H, 3D, 5A). This suggests that the pterygoids originally extended laterally in a trapezoidal and inserted into the palatine at the position corresponding to the posterior edge of the orbit (Figs. 1A, 1B, 2G, 2H and 3D).

In the previous description by *Lin & Rieppel (1998)* and *Holmes, Cheng & Wu (2008)*, the connection between the pterygoids and palatines on both sides was unclear (Fig. 3C). In European taxa such as *Neusticosaurus* (*Carroll & Gaskill, 1985*) and *Serpianosaurus* (*Rieppel, 1989*), the lateral margin of the pterygoid does not extend in trapezoidal form, while the lateral sides of the pterygoid of *Neusticosaurus peyeri* do not contact with the palatine (*Sander, 1989*). Similar lateral extensions of the pterygoid are present in *Wumengosaurus* (*Wu et al., 2011*), *Panzhousaurus* (*Lin et al., 2021*), *Diandongsaurus* (*Sato et al., 2014*; *Liu et al., 2021*) and *Dianmeisaurus* (*Shang & Li, 2015*), but these extensions are not trapezoidal. These observations suggest that the connection mode between pterygoid and palatine in *Keichousaurus* may represent an autapomorphy (Fig. 5).

## Ectopterygoid

The ectopterygoid is "L" shaped in ventral view, extending from the ventral side of the postorbital and curving anteroventrally to the dorsal edge of the dentary. The medial surface contacts the dorsal surface of the lateral region of the pterygoid, although the suture between them is indistinct (Fig. 3D). *Holmes, Cheng & Wu (2008)* described a similar bone located behind the palatine in several specimens, which was identified as the medial process of the maxilla instead of ectopterygoid. The "L" shaped ectopterygoid is similar to that of *Panzhousaurus rotundirostris* (*Lin et al., 2021*), *Diandongosaurus acutidentatus* (*Sato et al., 2014*) and *Dianmeisaurus gracilis* (*Shang & Li, 2015*). The ectopterygoid is also present in nothosaurs, such as *Nothosaurus rostellatus* (*Shang, 2006*) from Guizhou and *Simosaurus gaillardoti* (*Miguel Chaves, Ortega & Pérez-García, 2018*) from Europe, but in these taxa, the ectopterygoids are wide and flaky-like, contrasting sharply with the slender, curved "L" shaped ectopterygoid described in *Keichousaurus*.

## Hyobranchium

The hyobranchium is well preserved and rod-like in CUGW VH009 (Fig. 1B). The pair of hyobranchia are slightly curved along the shafts. They are slender and expanded at both the proximal and distal ends, and the proximal end is more pronounced than the distal end. The total length of the hyobranchial is approximately half the length of the orbit (Figs. 1 and 3D). Notably, the expansion of the distal end of the hyobranchium described by *Holmes, Cheng & Wu (2008)* appears to be the part of the posterior quadrate branch of the pterygoid rather than the hyobranchium (Figs. 1, 2C, 2D, 2G, 2H and 3C). The hyobranchium has been previously reported in *Serpianosaurus* (*Rieppel, 1989*), *Neusticosaurus* (*Carroll & Gaskill, 1985*; *Sander, 1989*), *Dactylosaurus* (*Sues & Carroll, 1985*), *Wumengosaurus* (*Wu et al., 2011*), *Dianmeisaurus* (*Shang & Li, 2015*) and *Diandongsaurus* (*Liu et al., 2021*), but remains unknown in *Dianopachysaurus* (*Liu et al., 2011*). The hyobranchium of *Keichousaurus* is more slender, compared to these taxa, with a hyobranchial-to-orbital length ratio similar to that of *Dianmeisaurus* (*Shang & Li, 2015*) and *Diandongsaurus* (*Liu et al., 2021*), but is larger than those of *Serpianosaurus* (*Rieppel, 1989*), *Neusticosaurus* (*Sander, 1989*) and *Dactylosaurus* (*Sues & Carroll, 1985*).

## Dentition

The premaxilla, maxilla and dentary teeth are all well preserved (Fig. 6). There are five premaxillary teeth as described by *Lin & Rieppel (1998)* and *Liao et al. (2021)*. The premaxillary teeth extend anteriorly, with the tips curving downward and posteriorly. They are generally larger than the maxillary teeth, and their size increases posteriorly, reaching the largest size in the fourth tooth. The fifth tooth is slightly smaller than the preceding four teeth (Fig. 6). In CUGW VH007 and CUGW VH009, which represent adult individuals of different sexes, the number of teeth on the right premaxilla increases gradually from one to four. The crown height of the fourth tooth of CUGW VH007 and CUGW VH009 reaches 1.2 mm and 1.4 mm, respectively. The crown height of the fifth tooth decreases to approximately 0.7 mm in CUGW VH007 and 0.9 mm in CUGW VH009. The left premaxillary teeth of CUGW VH007 and CUGW VH009 follow this pattern, but the

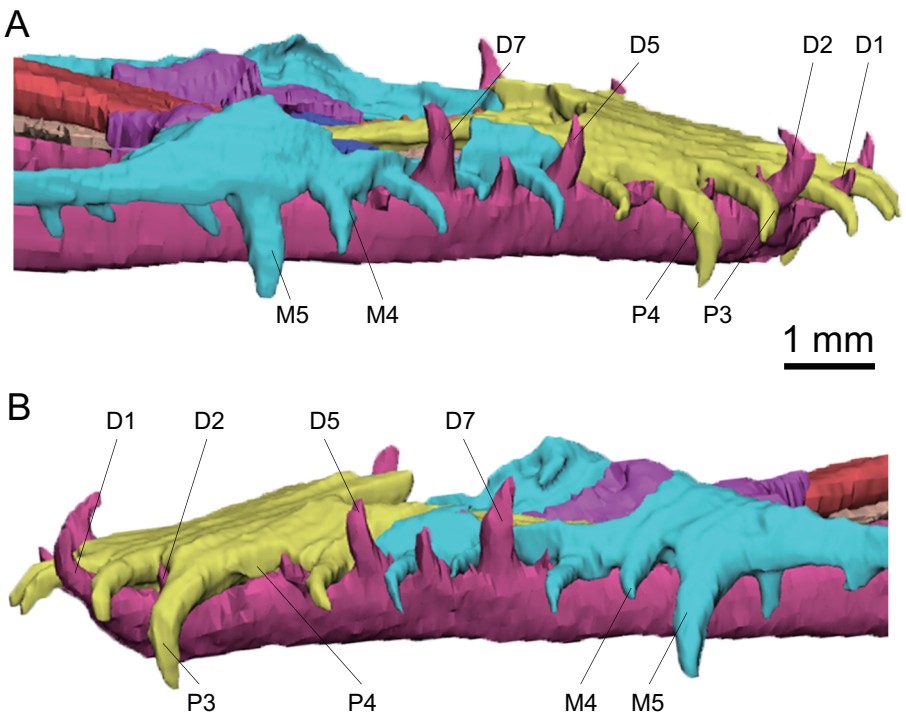

**Figure 6** **3D reconstruction of anterior teeth of *Keichousaurus hui* (CUGW VH009).** (A) Right side view; (B) left side view. The third tooth on the left dentary is absent. Abbreviations: D, dentary teeth; M, maxillary teeth; P, premaxillary teeth.

third and fourth teeth are rarely exposed, respectively (Figs. 2A, 2B and 6), which may be attributed to tooth replacement. However, CUGW VH017 exhibits no significant variation in tooth size, with the fourth tooth being the largest of the premaxillary teeth.

The anterior maxillary teeth (1–5) of *Keichousaurus* are well preserved, while the posterior teeth (6–15) are either missing or concealed by the maxilla. It can be inferred that there are 15 teeth on each side based on CT reconstruction, consistent with the description by *Holmes, Cheng & Wu (2008)*, and similar to that of *Liao et al. (2021)*. In all three specimens, the anterior maxillary teeth are slightly inclined, with the tips directed downward. The first three teeth are relatively small, conical in shape. In the adult CUGW VH007, the fourth and fifth teeth are widened. The worn crowns of the fourth and fifth teeth measure only 0.8 mm and 1.0 mm in height, but their anteroposterior length exceeds 0.4 mm, and their transverse width measures 0.3 mm and 0.4 mm, respectively, making them significantly thicker than the anterior three teeth. In another adult, CUGW VH009, the fourth tooth does not present a canine morphology. The fifth tooth, with a crown height of approximately 1.4 mm, and an anteroposterior length and a transverse width greater than 0.4 mm, is identified as a canine (Fig. 6). In CUGW VH017, the fourth teeth are poorly preserved, and the crowns of the fifth teeth are worn. The crowns of the left and right fifth teeth measure 0.6 mm and 0.7 mm in height, but the anteroposterior length and transverse width are both 0.3 mm, which are significantly larger than the anterior teeth. These are considered as canines

and might continue to grow. Variations in tooth size across different individuals may result from tooth replacement and preservation. However, the fourth and fifth teeth of the maxilla of *Keichousaurus* are described as canines by *Holmes, Cheng & Wu (2008)* and *Liao et al. (2021)*. From the sixth maxillary tooth, the teeth become smaller and more conical in shape, with a cone-shaped crown that forms a 60° angle with the horizontal. Canines in the maxilla have been reported in *Dianopachysaurus* (*Liu et al., 2011*), *Dawazisaurus* (*Cheng et al., 2016*), *Diandongosaurus* (*Sato et al., 2014*; *Liu et al., 2021*), *Dianmeisaurus* (*Shang & Li, 2015*). In *Dianopachysaurus*, the fourth of the six teeth in the right maxilla was described as significantly enlarged (*Liu et al., 2011*). In *Dawazisaurus*, the eighth of the twenty teeth was described as the largest of maxillary teeth and fang-like (*Cheng et al., 2016*). Five teeth were found in the anterior part of the maxilla of *Diandongosaurus*, and the third and fourth teeth were identified as canines. However, according to *Sato et al. (2014)* and *Liu et al. (2021)*, the space between the anterior edge of canines and the premaxilla can accommodate one or two additional teeth. In *Diandongosaurus*, the canines on the maxilla may be located in the position of the fourth and fifth teeth, which is similar to the canines in *Keichousaurus* (*Sato et al., 2014*). In *Dianmeisaurus*, at least four canines are present on the premaxilla, with the third tooth on the maxilla identified as a canine, which is slightly different from *Keichousaurus*. The shapes of canines and conical teeth are similar to *Keichousaurus* (*Shang & Li, 2015*).

In CUGW VH009, the anterior eight dentary teeth are well preserved, with a generally supine orientation and recurved tips. The dentary teeth are staggered with the premaxillary teeth and the maxillary teeth. The first, second, fifth and seventh teeth are larger than other dentary teeth, with crown heights ranging from 0.8 mm to 1.1 mm. The posterior dentary teeth (starting from the ninth tooth) are obscured by the maxilla. However, some very small teeth are visible through the orbit. A total of 21 dentary teeth are observed based on CT scanning images.

Compared with the traditional European pachypleurosaurs, the number of premaxillary teeth in *Keichousaurus* is similar to that of *Neusticosaurus pusillus* (5) and *Neusticosaurus peyeri* (5–6), but slightly fewer than that of *Neusticosaurus edwardsi* (6) and *Serpianosaurus mirigiolensis* (6–8). Compared with the Chinese pachypleurosaurs-like forms, the number of premaxillary teeth of *Keichousaurus* is the same as *Dawazisaurus brevis* (5), and similar to *Panzhousaurus rotundirostris* (at least 5 premaxillary teeth), *Dianopachysaurus dingi* (at least 5 premaxillary teeth), *Diandongosaurus acutidentatus* (at least 5 premaxillary teeth) and *Dianmeisaurus gracilis* (at least 4 premaxillary teeth), but slightly fewer than *Qianxisaurus chajiangensis* (8), and significantly fewer than *Honghesaurus longicaudalis* (10 premaxillary teeth perhaps) and *Wumengosaurus delicatomandibularis* (38).

Compared to traditional European pachypleurosaurs, the number of maxillary teeth in *Keichousaurus* is similar to that of *Serpianosaurus mirigiolensis* (15–16), but slightly more than *Neusticosaurus peyeri* (10–12) and *Neusticosaurus pusillus* (12), and possibly fewer than *Neusticosaurus edwardsi* (19 or more). The number of the maxillary teeth of Chinese pachypleurosaurs-like forms such as *Wumengosaurus delicatomandibularis*, *Panzhousaurus rotundirostris*, *Dianopachysaurus dingi* and *Diandongosaurus acutidentatus* is unknown. However, *Keichousaurus* has significantly fewer maxillary teeth than

*Wumengosaurus delicatomandibularis*, slightly fewer than *Honghesaurus longicaudali* (17–18) and *Dawazisaurus brevis* (20), the same as *Qianxisaurus chajiangensis* (15), and possibly slightly more than *Dianmeisaurus gracilis* (at least 13 teeth).

The number of dentary teeth in *Keichousaurus* is fewer than that of *Wumengosaurus delicatomandibularis* (at least 65 dentary teeth) and traditional European pachypleurosaurs, such as *Neusticosaurus peyeri* (24), *Neusticosaurus pusillus* (25) and *Serpianosaurus mirigiolensis* (31∼32). A detailed comparison of tooth morphology between *Keichousaurus*, other pachypleurosaur-like forms and pachypleurosaurs is provided in Table 2.

**Phylogenetic analysis**

This study uses the data matrix of *Wang et al. (2022)* which is mainly based on the data of *Neenan, Klein & Scheyer (2013)* with the combination of information from other researchers including *Liu et al. (2011)*. Six characters of *Keichousaurus* have been modified (Table S1, data from *Wang et al., 2022*).

The new matrix containing 181 characters and 63 genera, was analyzed by phylogenetic software TNT (*Goloboff, Farris & Nixon, 2008*). All characters are equally weighted and unordered. Phylogenetic analysis was conducted using a traditional search method with 100 replications of Wagner trees, one random seed and 10 trees saved per run. A total of 351 most parsimonious trees (TL = 811, CI = 0.293, RI = 0.685) were obtained in this analysis (Fig. S3).

In the strict consensus tree, the clades within Sauropterygia are poorly resolved. Five taxa were excluded from the reduced strict consensus tree (Fig. 7), including *Majiashanosaurus*, *Eremtmorhipis*, *Palatodonta*, *Hanosaurus*, and *Chaohusaurus*. The European pachypleurosaurs *Neusticosaurus*, *Serpianosaurus*, *Dactylosaurus*, *Odoiporosaurus*, and *Anarosurua*, form a monophyletic group, but the Pachypleurosauria from China do not form a monophyletic group. *Qianxisaurus* and *Wumengosaurus* are assigned to be Eosauropterygia. They form a polytomy with the European pachypleurosaurs and a branch that consists of the remaining genera of Eosauropterygia. On the contrary, *Dawazisaurus*, *Dianopachysaurus*, *Dianmeisaurus*, *Diandongosaurus*, and *Keichousaurus* form a monophyletic group with the remaining genera of European Eosauropterygia excluding *Panzhousaurus*. *Dawazisaurus*, *Dianopachysaurus*, *Dianmeisaurus*, and *Diandongosaurus* form a polytomy with a monophyletic group composed of *Keichousaurus*, *Plaudidraco*, *Simosaurus*, *Wangosaurus*, *Germansaurus* and two branches which are Nothosauridae and Pistosauroidea. This branch, which includes the Chinese pachypleurosaurs-like taxa except for *Qianxisaurus* and *Wumengosaurus* shared three derived characters: fully fused parietals in adults; a dorsal vertebrae count of 16–20; blunt and expanded ungual phalanges in pes that are wider than the proximal phalanges. The branch containing *Keichousaurus* and other genera of eosauropterygians is characterized by unambiguous synapomorphies: reduced nasals; three tarsal ossifications; short, blunt ungual phalanges in the pes that are not expanded.

## DISCUSSION

The CT scanning and skull reconstruction of three *Keichousaurus* have permitted us to make refinements and supplements in cranial anatomy and revealed previously unknown

**Table 2   Comparison of tooth characteristics of *Keichousaurus hui* with other pachypleurosaurs.**

| | Premaxillary teeth | Maxillary teeth | Dentary teeth |
|---|---|---|---|
| *Keichousaurus hui* (CUGW VH007, VH009, VH017) | 5 teeth are supine with the tip down, and the tips of the teeth tend to bend backward. One to four of them gradually become larger and the fifth one reduced in size. | 15 in total. The fourth and fifth are canines. The anterior teeth are also slightly supine. From the sixth, it suddenly becomes smaller, slightly extending anterioventrally, and is about 60 degrees to the horizontal level. | Among 21 teeth. The first, third, fifth and seventh teeth were relatively large, and all of them were supine with the tip upward. |
| *Serpianosaurus mirigiolensis* (*Rieppel, 1989*) | 6–8 teeth. The front teeth are larger and obviously curved medially, and the back teeth are slightly smaller than the maxillary teeth. | 15–16 teeth. | 31–32 teeth that are similar to the premaxillary teeth. The anterior teeth are larger and curved medially, and the posterior teeth were smaller, but all larger than the maxillary teeth. |
| *Neusticosaurus pusillus* (*Sander, 1989*) | 5 teeth similar in size (about 1.5 mm), with longitudinal ornamentation on the surface. | 12 teeth. Anterior teeth are large, similar to canines. These teeth are pointed and curved toward the tip, with longitudinal ornamentation. Their length is generally less than 1.5 mm, but it can reach 2 mm in adult. | 25 teeth. The tooth row of the dentary reaches further back than that of the maxilla. |
| *Neusticosaurus peyeri* (*Sander, 1989*) | 5–6 teeth | 10–12 teeth | 24 teeth. Some teeth are larger. |
| *Neusticosaurus edwardsi* (*Carroll & Gaskill, 1985*) | 6 teeth, large and supine. | 19 teeth or more; the front teeth are as large and supine as the premaxillary teeth, and the back teeth become smaller and straight. | Unknown |
| *Panzhousaurus rotundirostris* (*Jiang et al., 2018*) | At least 5 teeth, maybe 6 teeth in total. Recurved and procumbent. | 8 teeth are preserved, and the description is same as premaxillary. The last two preserved teeth are robust. | Unknown |
| *Wumengosaurus delicatomandibularis* (*Jiang et al., 2008*; *Wu et al., 2011*) | 38 teeth, small, monocuspid, and vertically positioned, with striations, and the tooth crown expands basally, basal pedicel is constricted. | More than 50 teeth, and they are similar to premaxillary teeth. | At least 65 teeth, and they are similar to premaxillary teeth. |
| *Diandongosaurus acutidentatus* (*Sato et al., 2014*; *Liu et al., 2021*) | At least 5 teeth, large and supine, similar to canines. | 6 or 7 maxillary teeth that are large and supine. The fourth and fifth teeth are canines. | The anterior teeth are similar to the premaxillary teeth, but the posterior teeth are not preserved. |
| *Dianopachysaurus dingi* (*Liu et al., 2011*) | At least 5 teeth, slightly supine, with the most anterior teeth bent inward. | 6 teeth. The fourth increased significantly, and the rest of the teeth are smaller than the premaxillary teeth. The last one is below the posterior margin of the orbit. | Unknown |

**Table 2** (*continued*)

| | Premaxillary teeth | Maxillary teeth | Dentary teeth |
|---|---|---|---|
| *Dianmeisaurus gracilis* (*Shang & Li, 2015*) | At least 4 canines. The crown of canines is slightly curved dagger shaped, with longitudinal ridges on the surface. | At least 13 teeth with one canine, two maxillary teeth in front of canine and more than 10 teeth behind the canine. | The front canines correspond to the premaxillary teeth, and the rear teeth are small conical teeth. The posterior edge of the dentition is located in the anterior part of the posterior edge of the orbit. |
| *Dawazisaurus brevis* (*Cheng et al., 2016*) | 5 teeth, sub-conical and curve medially and slightly posteriorly, with fine striations on the crown surface. The fourth and fifth are the largest and the smallest, respectively. The fourth is fang-like. | 20 teeth in total, and the description is same as premaxillary. The shape of eighth just like the fourth of premaxillary, but slightly smaller than latter, while others are much smaller and have similar size. | Unknown |
| *Honghesaurus longicaudalis* (*Xu et al., 2022*) | 10 teeth are estimated, and 8 teeth are preserved and other gaps for 2 teeth are missed in the holotype. They are homodont with a tall peduncle, a short and conical crown, and smooth lateral surface. | 17–18 teeth are estimated, when 12 teeth are discernable and 5 or 6 teeth are missing. They are larger than exhibit same morphology with the premaxillary teeth. | Unknown |
| *Qianxisaurus chajiangensis* (*Cheng et al., 2012*) | 8 teeth are incompletely preserved in the holotype, when the anterior teeth are slender and the posterior ones become gradually more robust. | 15 maxillary teeth with the first two and the last two are obviously small. They possess a tall, slightly constricted peduncle and a short, conical crown bearing many fine striations on the lateral surface. | Unknown |

features. In this study, *Keichousaurus* are compared with the Triassic eosauropterygians *Anarosaurus*, *Dactylosaurus*, *Serpianosaurus*, *Neusticosaurus* from Europe (*Carroll & Gaskill, 1985*; *Sues & Carroll, 1985*; *Sander, 1989*; *Rieppel, 1989*; *Rieppel & Lin, 1995*; *Klein, 2009*; *Renesto, Binelli & Hagdorn, 2014*) and *Qianxisaurus*, *Wumengosaurus*, *Honghesaurus*, *Panzhousaurus*, *Dianopachysaurus*, *Diandongosaurus*, *Dianmeisaurus*, *Dawazisaurus*, *Majiashanosaurus* from South China (*Jiang et al., 2008*; *Jiang et al., 2014*; *Jiang et al., 2018*; *Shang, Wu & Li, 2011*; *Wu et al., 2011*; *Liu et al., 2011*; *Liu et al., 2015*; *Cheng et al., 2012*; *Cheng et al., 2016*; *Sato et al., 2014*; *Shang & Li, 2015*; *Lin et al., 2021*; *Hu, Li & Liu, 2024*).

Generally, *Keichousaurus* shares more similarities with the basal Chinese eosauropterygians *Qianxisaurus*, *Panzhousaurus*, *Dianopachysaurus*, *Diandongosaurus*, *Dianmeisaurus* and *Dawazisaurus*, than the Chinese eosauropterygians *Wumengosaurus* and *Honghesaurus*, as well as the European pachypleurosaurs *Anarosaurus*, *Serpianosaurus*, *Neuticosaurus* and *Dactylosaurus*. These similarities include laterally extending protrusions on the lateral side of the pterygoid, "L" shaped ectopterygoid, and the number of premaxillary teeth. Among these taxa, *Keichousaurus* is particularly similar to *Diandongosaurus* and *Dianmeisaurus*.

In previous phylogenetic studies, the relationship between *Keichousaurus*, European pachypleurosaurs and other similar Chinese forms has been controversial (*Li & Liu, 2019*). The phylogenetic relationships among Pachypleurosaurs, as well as eosauropterygians
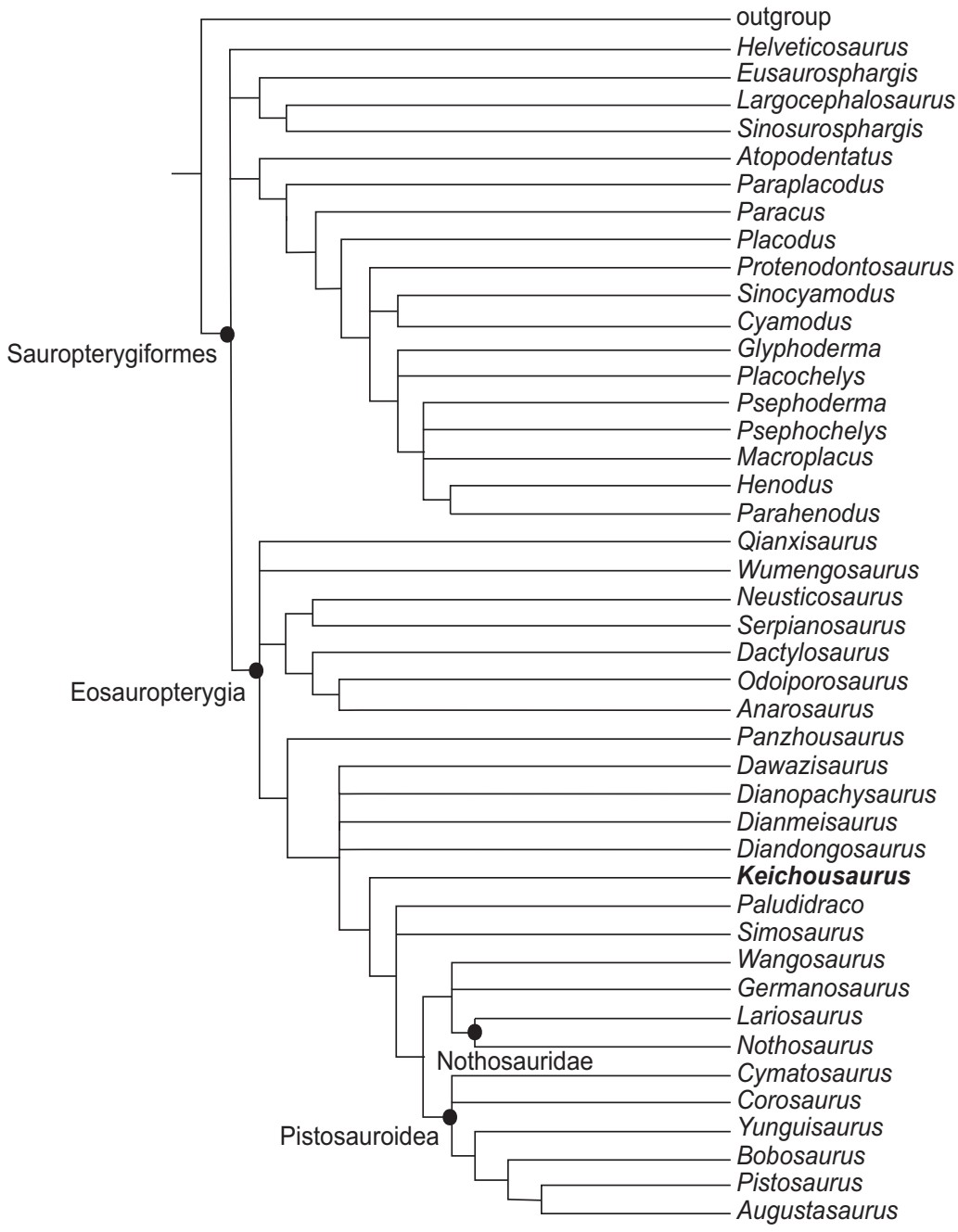

**Figure 7** **Reduced strict consensus tree of 351 MPTs showing the phylogenetic relationships of** ***Keichousaurus* in Sauropterygia.** Five taxa excluded, including *Majiashanosaurus*, *Eremtmorhipis*, *Palatodonta*, *Hanosaurus*, and *Chaohusaurus*. TL = 811, CI = 0.293, RI = 0.685.

in general, remain unresolved. In this study, we discuss the relationship between *Keichousaurus* and three aforementioned taxa with other families of Eosauropterygia. European pachypleurosaurs including *Anarosaurus*, *Serpianosaurus*, *Neuticosaurus,* and

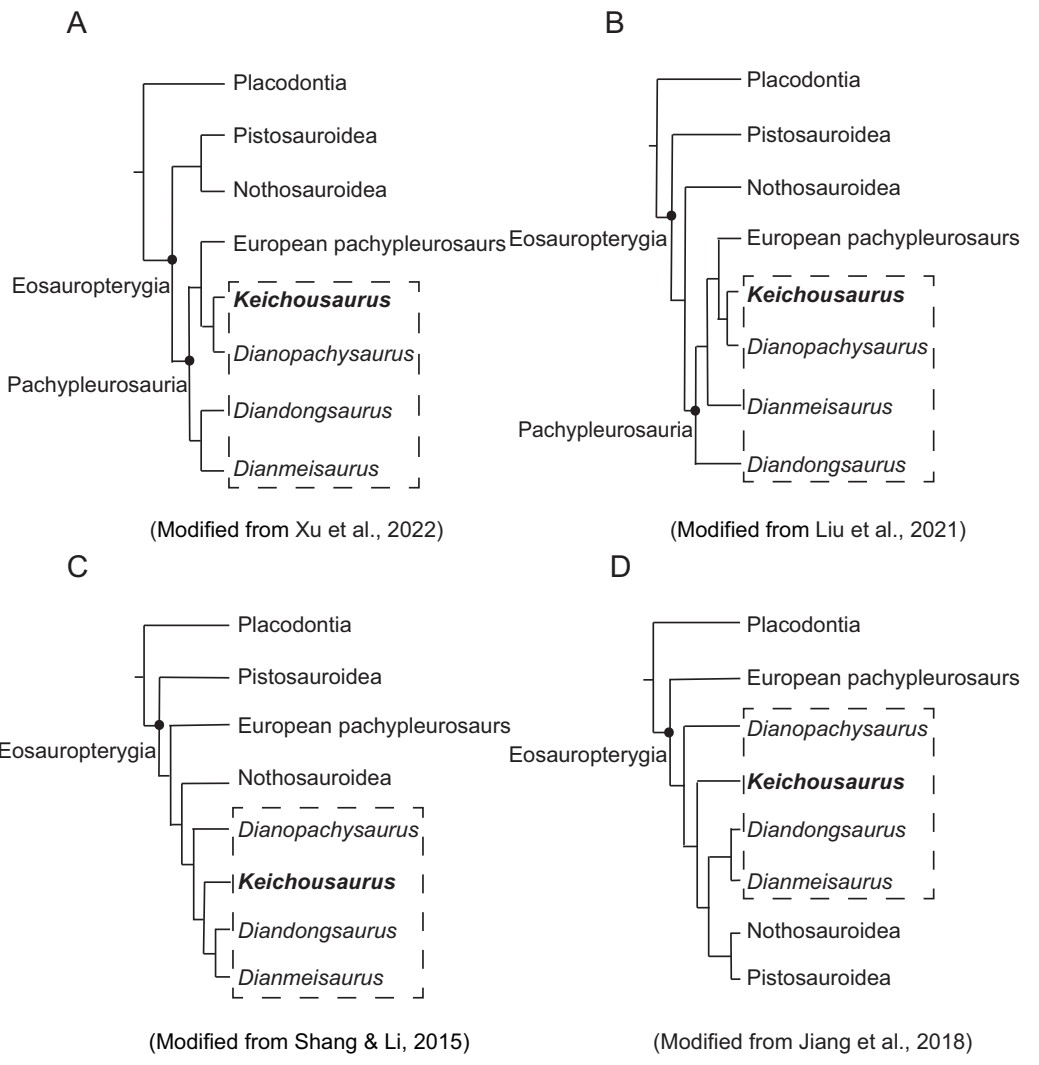

**Figure 8 Phylogenetic cladograms showing the relationship of Sauropterygia.** The dotted box shows taxa of Chinese pachypleurosaurs or pachypleuosaur-like forms.

*Dactylosaurus* display a stable interrelationship, and constantly form a monophyletic group in most studies.

Some studies support the monophyly of Pachypleurosauria, but there are differences in the internal relationships within Eosauropterygia and the position of the Chinese genera (Figs. 8A and 8B). Regarding the internal relationship of Eosauropterygian families, some analyses place Nothosauroidea and Pistosauroidea in a monophyletic group, forming a sister clade to Pachypleurosauria (*Liu et al., 2011*; *Liu et al., 2021*; *Xu et al., 2022*) (Fig. 8A). Other studies suggest that Pistosauroidea is the basal calde of the consecutive sister groups of Nothosauroidea and Pachypleurosauria (*Neenan, Klein & Scheyer, 2013*; *Li & Liu, 2020*; *Liu et al., 2021*; *Hu, Li & Liu, 2024*) (Fig. 8B). In these analyses, the positions of Chinese

pachypleurosaurs or pachypleuosaur-like forms are unstable, although *Keichousaurus* consistently appears closer to European pachypleurosaurs (Figs. 8A and 8B).

The monophyly of Pachypleurosauria is invalid in some studies (Figs. 8C and 8D). In Fig. 8C, Pistosauroidea is positioned at the base of Eosauropterygia. The European pachypleurosaurs form a monophyletic group with a branch composed of Nothosauroidea and Chinese pachypleurosaurs or similar forms and become the sister group to Pistosauroidea. This suggests that the four taxa of Chinese pachypleurosaurs-like forms are more derived than others, and *Keichousaurus* is more closely related to Nothosauroidea than to European pachypleurosaurs (Fig. 8C). This relationship between *Keichousaurus*, Nothosauroidea and European pachypleurosaurs has also been supported by some authors (*Holmes, Cheng & Wu, 2008*; *Shang, Wu & Li, 2011*; *Wu et al., 2011*). In contrast, the phylogenetic analysis of *Ma et al. (2015)* and *Jiang et al. (2018)* places Nothosauroidea and Pistosauroidea as sister groups, forming a monophyletic clade that is sister to the monophyletic group of *Diandongsaurus* and *Dianmeisaurus*. This larger clade and other Chinese pachypleurosaurs-like taxa form a monophyletic group, and this big group is a sister group to European pachypleurosaurs. *Keichousaurus* is more derived than *Dianophachysaurus* (*Ma et al., 2015*; *Jiang et al., 2018*) (Fig. 8D).

Compared to previous studies, the result of our phylogenetic analysis is similar to that of *Jiang et al.* (*2018*; Fig. 8D), though the position of *Keichousaurus* is more derived than *Diandongosaurus* and *Dianmeisaurus*. In our analysis, *Keichousaurus* is basal to the clade including Nothosauridae and Pistosauroidea. However, the support for the clade including *Keichousaurus*, Nothosauridae and Pistosauroidea is generally low, which suggests this clade is not stable (Fig. S3). Therefore, further research is necessary to explore the characters of *Keichousaurus* within the Eosauropterygia.

## CONCLUSION

In this study, new cranial anatomy data for three specimens of *Keichousaurus* were provided through CT scanning. These include the L-shaped ectopterygoid, the wedge-shaped posterolateral process of the frontal, the trapezoidal pterygoid for articulating with the palatine, and the rodlike basioccipital tuber. The phylogenetic analysis using the revised matrix with new features suggests that *Keichousaurus* is more closely related to derived Chinese pachypleurosaurs-like within Eosauropterygia. Additionally, *Keichousaurus* may be more derived than other Chinese pachypleurosaurs-like forms, which are positioned basal to the clade that includes Nothosauridae and Pistosauroidea.

## ACKNOWLEDGEMENTS

We thank Feng Yun from the Institute of Vertebrate Paleontology and Paleoanthropology, Chinese Academy of Sciences and Yinghua NDT (Shanghai) Co., Ltd. for CT scanning of fossils, thank Jinfeng Hu from China University of Geoscience (Wuhan) for reviewing the CT scans, thank Rui Wu and Han Yao from China University of Geoscience (Wuhan) for editing this manuscript. We also thank the editor Alexander Ereskovsky and the reviewers Carlos de Miguel Chaves and Melani Berrocal Casero for their very useful comments.

### Funding

This project is supported by the National Natural Science Foundation of China (41530104, 42288201, 42002019, and 41972014) and China Geological Survey (DD20230006). The funders had no role in study design, data collection and analysis, decision to publish, or preparation of the manuscript.

### Grant Disclosures

The following grant information was disclosed by the authors:
National Natural Science Foundation of China: 41530104, 42288201, 42002019, 41972014.
China Geological Survey: DD20230006.

### Competing Interests

The authors declare there are no competing interests.

### Author Contributions

- Jiayu Xu conceived and designed the experiments, performed the experiments, analyzed the data, prepared figures and/or tables, authored or reviewed drafts of the article, and approved the final draft.
- Yu Guo conceived and designed the experiments, performed the experiments, prepared figures and/or tables, authored or reviewed drafts of the article, and approved the final draft.
- Yucong Ma conceived and designed the experiments, performed the experiments, analyzed the data, prepared figures and/or tables, authored or reviewed drafts of the article, and approved the final draft.
- Wei Wang performed the experiments, authored or reviewed drafts of the article, and approved the final draft.
- Long Cheng performed the experiments, authored or reviewed drafts of the article, and approved the final draft.
- Fenglu Han conceived and designed the experiments, performed the experiments, prepared figures and/or tables, authored or reviewed drafts of the article, and approved the final draft.

### Data Availability

The photographs and the schematic map of specimens, the code of Keichousaurus in the matrix, and strict consensus tree are available in the Supplementary Files.

The reconstructions of the specimens CUGW VH007, CUGW VH009 and CUGW VH017 are available at figshare and MorphoSource:

- CUGW VH007, https://doi.org/10.6084/m9.figshare.26947846.v1; https://doi.org/10.17602/M2/M665692.
- CUCW VH009, https://doi.org/10.6084/m9.figshare.26947537.v2; https://doi.org/10.17602/M2/M665698.

- CUGW VH017, https://doi.org/10.6084/m9.figshare.26947852.v1; https://doi.org/10.17602/M2/M665695.

## Supplemental Information

Supplemental information for this article can be found online at http://dx.doi.org/10.7717/peerj.19012#supplemental-information.

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
