# Peer review of "New digital anatomical data of Keichousaurus hui (Reptilia: Sauropterygia) and its phylogenetic implication"

_PeerJ, doi:10.7717/peerj.19012_

## Round 0.1 · original submission · Minor Revisions

· Academic Editor

Minor Revisions

Dear Dr. Xu,

Please pay special attention to the comment of the first reviewer who asks that the anatomical comparisons could include two or three more taxa within the early Sauropterygia clade.

Best regards,
Editor Alexander Ereskovsky

·

Basic reporting

This is an interesting, precise and concise manuscript about the cranial anatomy of the early sauropterygian Keichousaurus. The use of of CT-scans has provided new details about the anatomy of this taxon, emending some erroneous previous asumptions as well as providing novel information.These findings are incorporated in a data matrix, resulting in new phylogenetic analysis. In addition, the manuscript is supported by an important number of detailed figured.

My main concern is that the lenguage and grammar could be improved. Although English is not my native language, i find some sentences a little bit confusing, quite unclear, or informal at some points. Maybe a language review by a native speaker would be useful.

Other minor concern is that the anatomical comparisons could include two or three more taxa within the early Sauropterygia clade, although I think this may be justified if the authors couldn't get access to those specimens.

Experimental design

The CT-scans and descriptions of the new elements can be replicable and checked by other scientists, resulting in an improvement of the knowledge on the cranial anatomy of this taxon. The research is focused and well defined.

Validity of the findings

The discussion of the results is valid, the phylogenetic analysis and descriptions of the elements being well compared with previous studies on these taxon, as well as other early sauropterygian taxon. This information can be incoporated in the future for new studies on the phylogeny of Triassic sauropterygians, being the discussed and problematic monophily of pachypleurosaurs an interesting topic within this field.

Additional comments

Attached the editors and authors will find a pdf with some additional comments and minor corrections. as stated before, I consider this manuscript as publishable after some minor-moderate revisions, mostly related to the language and grammar.

·

Basic reporting

The relationships among eosauropterygians are frequently debated and remain without consensus to date. This paper presents relevant findings that improve our understanding of the anatomy of Keichousaurus and its possible position within Eosauropterygia.However, the text must be improved.

There are some suggestions for improving the English in the attached PDF. Sometimes, it is better to use the past tense or the present continuous instead of the present simple (abstract).

IMPORTANT. The genus and species names should always be in italics. Please review, as I noticed they were not italicized in the table 2 and in some references (ex. Kueichosaurus hui in line 402..., line 445, 490, 525, 541, 545, Simosaurus and Nothosaurus in lines 498...)
Some references are missing.

To cite a reference with more than two authors, use the surname of the first author followed by "et al." + year; please review all the text.

Revise the text in lines 242-254. Some ";" must be changed by . (point).

The style of some parts of the text must be imporved (specially "Conclusions"). The use of the first person "we" is not "formal" in scientific language.


SEE ATTACHED PDF for more detailed comments.

Sincerely,
Dr. Mélani Berrocal-Casero
Associate Professor (Profesora Ayudante Doctora) University of Alcalá (UAH; Madrid, Spain)

Experimental design

No comment.

Validity of the findings

No comment

Additional comments

See attached PDF

---

## Round 0.2 · accepted · Accept

· Academic Editor

Accept

Dear Dr. Xu,

Thank you for your careful work in revising the manuscript in accordance with the recommendations of the reviewers. In this form, your manuscript is ready for publication.

Sincerely,
Alexander Ereskovsky